# Mating-Type Locus Organization and Mating-Type Chromosome Differentiation in the Bipolar Edible Button Mushroom *Agaricus bisporus*

**DOI:** 10.3390/genes12071079

**Published:** 2021-07-16

**Authors:** Marie Foulongne-Oriol, Ozgur Taskent, Ursula Kües, Anton S. M. Sonnenberg, Arend F. van Peer, Tatiana Giraud

**Affiliations:** 1INRAE, MycSA, Mycologie et Sécurité des Aliments, 33882 Villenave d’Ornon, France; marie.foulongne-oriol@inra.fr; 2Ecologie Systématique Evolution, Bâtiment 360, CNRS, AgroParisTech, Université Paris-Saclay, 91400 Orsay, France; ozgur.taskent@universite-paris-saclay.fr; 3Molecular Wood Biotechnology and Technical Mycology, Goettingen Center for Molecular Biosciences (GZMB), Büsgen-Institute, University of Goettingen, Büsgenweg 2, 37077 Goettingen, Germany; ukuees@gwdg.de; 4Plant Breeding, Wageningen University and Research, Droevendaalsesteeg 1, 6708 PB Wageningen, The Netherlands; anton.sonnenberg@wur.nl (A.S.M.S.); arend.vanpeer@wur.nl (A.F.v.P.)

**Keywords:** mating-type chromosome, mating-type genes, bipolar, tetrapolar, pseudo-homothallic, heterothallic, automixis, *Agaricus*, basidiomycete, genetic map, homeodomain genes, pheromone genes, pheromone receptor genes, recombination suppression, sex chromosomes

## Abstract

In heterothallic basidiomycete fungi, sexual compatibility is restricted by mating types, typically controlled by two loci: *PR*, encoding pheromone precursors and pheromone receptors, and *HD*, encoding two types of homeodomain transcription factors. We analysed the single mating-type locus of the commercial button mushroom variety, *Agaricus bisporus* var. *bisporus*, and of the related variety *burnettii*. We identified the location of the mating-type locus using genetic map and genome information, corresponding to the *HD* locus, the *PR* locus having lost its mating-type role. We found the *mip1* and *β-fg* genes flanking the *HD* genes as in several Agaricomycetes, two copies of the *β-fg* gene, an additional *HD2* copy in the reference genome of *A. bisporus* var. *bisporus* and an additional *HD1* copy in the reference genome of *A. bisporus* var. *burnettii.* We detected a 140 kb-long inversion between mating types in an *A. bisporus* var. *burnettii* heterokaryon, trapping the *HD* genes, the *mip1* gene and fragments of additional genes. The two varieties had islands of transposable elements at the mating-type locus, spanning 35 kb in the *A. bisporus* var. *burnettii* reference genome. Linkage analyses showed a region with low recombination in the mating-type locus region in the *A. bisporus* var. *burnettii* variety. We found high differentiation between *β-fg* alleles in both varieties, indicating an ancient event of recombination suppression, followed more recently by a suppression of recombination at the *mip1* gene through the inversion in *A. bisporus* var. *burnettii* and a suppression of recombination across whole chromosomes in *A. bisporus* var. *bisporus*, constituting stepwise recombination suppression as in many other mating-type chromosomes and sex chromosomes.

## 1. Introduction

Mating systems are responsible for the degree of selfing/outcrossing in natural populations and impact gene flow, the accumulation of deleterious alleles, and adaptability [1,2,3,4,5,6]. Mating systems can be controlled by, and reciprocally influence, genetic systems governing mating compatibility, such as self-incompatibility systems in plants [7]. In heterothallic fungi, sexual compatibility is governed by a mating-type system, which is most often controlled by two loci segregating independently in basidiomycetes (e.g., rusts, smuts and mushrooms) [8]. To mate and produce offspring successfully, two haploid cells must carry different alleles at both loci. Four different mating types are thus produced following meiosis in the heterothallic fungi with two mating-type loci, and they are thus called tetrapolar. Multiple major transitions in basidiomycetes have led to a control by a single mating-type locus, either by linkage between the two loci via chromosomal rearrangement and recombination suppression or by one locus losing its role in mating-type determinism [8,9,10,11,12]. Two different mating types are produced following meiosis in the heterothallic fungi with one mating-type locus, and they are thus called bipolar. Carrying two mating-type loci with multiple alleles, such as in many mushrooms, favours outcrossing, while carrying a single mating-type locus is beneficial under selfing mating systems [9,13]. In basidiomycete fungi in which mating-type loci are linked by recombination suppression, further extension of the cessation of recombination has been reported beyond mating-type genes, in a stepwise manner, forming large non-recombining mating-type chromosomes displaying “evolutionary strata” of differentiation between mating-type chromosomes [10,14,15].

The *Agaricus* fungi are good models for studying mating-type genetic control and their relationship to mating systems because they exhibit a range of different mating systems and of genetic determinisms of mating-type [16,17,18,19,20,21]. Some species are heterothallic, forming haploid spores that are compatible for mating only with haploid cells carrying a different mating type, and these species mostly outcross. Other species are homothallic, i.e., there is no discrimination for mating based on any mating-type gene. Yet other species are pseudo-homothallic, i.e., they form spores that carry two nuclei, of different mating types, issued from a single meiosis; these therefore mostly self by automixis, i.e., mating among products of a single meiosis [16,17,18,19,20,21].

Cultivars of the commercial button mushroom variety, *A. bisporus* var. *bisporus*, are mostly pseudo-homothallic, while the closely related variety *A. bisporus* var. *burnettii* is mostly heterothallic [17]. Both have a single locus controlling mating type [22,23], thus constituting a good model for studying the evolution of breeding systems. Pseudo-homothallism is manifested in *A. bisporus* var. *bisporus* by two-spored basidia, whose spores contain two haploid non-sister nuclei from a single meiosis, i.e., two haploid post-meiotic nuclei of different mating types. Such types of heterokaryotic basidiospores (n + n) germinate into heterokaryotic mycelia that contain in their cells the two haploid nuclei of different mating types, albeit in variable numbers. Under appropriate environmental conditions, the fertile heterokaryons can form fruiting bodies with binucleate basidia in which the two different types of haploid nuclei undergo karyogamy and meiosis for subsequent basidiospore production [22,24,25,26]. Such karyogamy between nuclei from a single meiosis constitutes automixis, a.k.a. intra-tetrad mating. Pseudo-homothallism may have been selected for during *A. bisporus* var. *bisporus* domestication for the homogenisation of the genetic material and the conservation of favourable traits, as automixis largely maintains the parental genotype [27,28,29,30]. Furthermore, the non-sister segregation in meiosis, with recombination being restricted at the very edges of chromosomes in the studied cultivated strain of *A. bisporus* var. *bisporus*, ensures maintenance of heterozygosity across most loci [27,31,32,33]. While the formation of bi-spored basidia in the pseudo-homothallic *A. bisporus* var. *bisporus* is predominant (the spores being thus heterokaryotic, n + n), low percentages of tri- or four-spored basidia occur with two or four, respectively, homokaryotic spores that receive just a single haploid meiotic nucleus (n) [34,35,36]).

The four-spored heterothallic closely related variety *A. bisporus* var. *burnettii*, known from the Sonoran Desert of California [17,37], has on average 11 recombination events per individual per generation, evenly distributed along chromosomes [31,38]. As in the var. *bisporus*, mating of the variety *burnettii* is governed by a single mating-type locus [37,39]. The formation of bi- or tetrasporic basidia in the two varieties and in crossbreds has been suggested to be under the main control of a locus, *BSN*, on chromosome 1 [40,41]. In both varieties, the mating-type locus is also on chromosome 1 [23] and multiallelic with at least 14 different specificities [39].

In the standard life cycle of the tetrapolar basidiomycetes, mating type is controlled by two loci. One locus (the *HD* locus) encodes two types of homeodomain transcription factors (*HD1* and *HD2*), and the other (the *PR* locus) encodes precursors for lipid-modified pheromones and STE3-like pheromone receptors [8]. In many mushrooms, the *HD* locus is made up of several paralogous subloci, each containing a *HD1*-*HD2* pair [42]. The *PR* locus can also be composed of several paralogous subloci, each with pheromone precursor and pheromone receptor genes [43]. In the Tremellomycetes, Trichosporonales, Microbotryales and in the Ustilaginales, bipolarity has repeatedly evolved, through the linkage by recombination suppression of the two *MAT* loci with functional *HD* and functional *PR* genes into a single inherited unity [8,10,12,14,44]. Bipolarity has also repeatedly evolved in the Agaricomycetes from tetrapolar systems [45], however following the loss of the mating-type function at the *PR* locus, while the locus was retained in the genomes [46,47,48,49,50]. Loss of mating-type function may not mean full loss of function of the genes. Pheromone precursor and pheromone receptor genes might still exert functions in sexual reproduction such as in the process of fruiting as shown for genes of the bipolar *Coprinellus disseminatus* upon transformation into the heterologous host *Coprinopsis cinerea* [48].

From the published whole genome analysis of *A. bisporus* var. *bisporus* strain H97 and *A. bisporus* var. *burnettii* strain JB137-S8, the single *MAT* locus present on chromosome 1 [23] has been connected to *HD* genes [51]. Here, we identified the precise location of the mating-type locus on chromosome 1 using available genetic map and genome information in *A. bisporus* var. *burnettii* and *A. bisporus* var. *bisporus* and thus inferred that the mating-type locus corresponds to the *HD* locus. In the reference genomes H97 and JB137-S8, we found the same gene order *mip1*-*HD1* and *HD2* mating type genes-*β-fg* with an *HD1*-*HD2* gene pair divergently transcribed as in several other Agaricomycetes. We also found two copies of the *β-fg* gene (as noted in a preliminary *MAT* locus analysis [52]), an additional *HD2* copy in the reference genome of *A. bisporus* var. *bisporus* and an additional *HD1* copy in the reference genome of *A. bisporus* var. *burnettii* with a short 5′ fragment remnant of an *HD2* gene, that may be footprints of additional *HD* subloci. In addition, the two reference strains had islands of transposable elements inserted at the mating-type locus, but in different locations, spanning 35 kb in the JB137-S8 *A. bisporus* var. *burnettii* strain and 1.6 kb in the H97 *A. bisporus* var. *bisporus* strain. We found four genes encoding pheromone precursors, three of them localizing on chromosome 1, flanking the two ends of a large island rich in transposable elements, corresponding to the centromere, far from the mating-type locus. We detected four genes encoding putative STE3-like pheromone receptors scattered across the genome and unlinked to the mating-type locus; we identified the ortholog to the *PR* mating-type locus on chromosome 13, and we describe its structure. By analysing monokaryons isolated from heterokaryons (from the reference U1 strain for *A. bisporus* var. *bisporus* and from the 119/9 strain for *A. bisporus* var. *burnettii*), we found mosaic patterns of differentiation between mating-type chromosomes in both varieties, but distinct patterns in the two varieties and a 140 kb-long inversion between mating types in *A. bisporus* var. *burnettii*, trapping the *HD* genes, the *mip1* gene and fragments of other genes. Linkage analyses showed a region with low recombination in the mating-type locus region in the *A. bisporus* var. *burnettii* variety. This ancient event of recombination cessation was followed more recently by a suppression of recombination across whole chromosomes in *A. bisporus* var. *bisporus*, thus constituting a stepwise extension of recombination suppression (with two steps).

## 2. Material and Methods

### 2.1. Strains, Genomes and Progenies

Mating-type genes and other genes from homokaryons *A. bisporus* var. *bisporus* H97 (v2.0; https://mycocosm.jgi.doe.gov/Agabi_varbisH97_2/Agabi_varbisH97_2.home.html accessed on 27 March 2011) and *A. bisporus* var. *burnettii* JB137-S8 (https://mycocosm.jgi.doe.gov/Agabi_varbur_1/Agabi_varbur_1.home.html on 27 March 2011) were deduced from the established genome annotation versions of the DOE Joint Genome Institute (JGI; Walnut Creek, CA) by blast, blastp and/or tblastn searches using suitable sequences from NCBI GenBank (http://www.ncbi.nlm.nih.gov/ on 27 March 2011), respectively, sequences from the JGI genome pages of *C. cinerea* (http://genome.jgi-psf.org/Copci1/Copci1.home.html on 27 March 2011) and of the *Agaricus* strains, and by blast, blastp and/or tblastn searches using *Agaricus* sequences against the GenBank and the JGI *C. cinerea* site. H97 is one component homokaryon from the heterokaryon Horst U1 [53] used for EST sequencing and genome annotation [51].

The haploid progeny (referred hereafter as *H_i_*) used for the mapping purpose was derived from the intervarietal hybrid JB3-83 x U1-7 [17,38]. JB3-83 is issued from *A. bisporus* var. *burnetti* JB3 [17] that belongs to the same Sonoran desert population as JB137 (from which the sequenced strain JB137-S8 originates [51]), and U1-7 is identical to one of the constituent nuclei of Horst U1, i.e., homokaryon H39 [53]. The segregation of the *MAT* locus in the *H_i_* progeny has been described previously [41]. Details on the obtention of mapping data and on the construction of the genetic linkage map are available in [38]. In order to refine the location of the *MAT* locus, the segregation data of the molecular markers assigned to chromosome 1 [38,54] were recomputed using Mapmaker/exp V3.0b software [55]. Molecular markers of known sequences were assigned to the H97 genome sequence V3.0 by blastn searches. The *MAT* locus was then placed in the next round of the process of linkage mapping using available progeny data. Marker orders were tested using the *ripple* command. The multipoint likelihood of alternative orders was computed, and orders with log-likelihoods within same magnitude were considered as equally likely.

For differentiation patterns along chromosomes, we used for *A. bisporus* var. *bisporus* the genomes of H97 and H39, the constituent homokaryons with opposite mating types isolated from the cultivated heterokaryon U1. The two genomes of *A. bisporus* var. *burnetti* analysed, H119p1 and H119p4, are the two constituent homokaryons of opposite mating types isolated from the heterokaryon 119/9 (JB-13), originating from the Agaricus Recovery Program [31].

### 2.2. Analyses of Protein Products

For protein comparisons, sequences were either taken from NCBI GenBank or from the JGI Mycocosm databases and aligned using ClustalX ([56]) and GeneDoc version 2.6.002 [57]. Phylogenetic trees were built by the neighbor joining method (bootstrap 500 replicates) of MEGA version 5.0 [58]. The CFSSP: Chou & Fasman Secondary Structure Prediction Server ([59]), the TMpred server ([60]), the TMHMM Server v. 2.0 ([61]) and the SPLIT 4.0 Server ([62]) were used to predict helical structures and transmembrane regions, respectively.

### 2.3. Orthology Reconstruction and Synonymous Divergence

Short-read sequences of the genomes of the constituent homokaryons of the var. *burnettii* 119/9 strain (i.e., H119p1 and H119p4) [31] were assembled with SPAdes v3.11.1 [63,64] with a hybrid assembly approach by using the available PacBio genome of H119p4 [31] as a reference with the “--trusted-contigs” flag.

As gene models were not available for *A. bisporus* var. *bisporus* homokaryon H39 and *A. bisporus* var. *burnettii* 119/9 genomes (i.e., H119p1 and H119p4), *ab-initio* gene annotations for these three genomes were obtained by using protein sequences of the var. *bisporus* homokaryon H97 v3.1 [27] and var. *burnettii* strain JB137-S8 [51] with BRAKER v2.1.5 [65,66,67,68,69]. The *HD1* gene in H119.p1 remained undetected by automatic prediction but could be manually defined by its position in between the *mip* and the *HD2* gene annotations and in structure by blasting with other *A. bisporus* and *C. cinerea HD1* proteins. To find orthologous gene copies in the constituent homokaryon genomes of var. *bisporus* Horst U1 and var. *burnettii* 119/9, all-against-all BLAST searches were first performed for each pair of homokaryotic genomes with DIAMOND [67]. The output tsv files of DIAMOND were then used as the input files for orthAgogue [70] to find orthologous pairs of genes for each pair of homokaryotic genomes. Co-orthologous pairs of genes were aligned with macse v2.04 [71]. Synonymous substitution rates (dS) were estimated with the yn00 method of PAML version 4.9f [72]. Custom python and shell scripts were used to work with multiple loci simultaneously, and the plots were made with R version 4.0.3 [73].

As the hybrid assemblies of the H119p1 and H119p4 genomes were highly fragmented, and thus were not assembled into complete chromosomes, the full gene order for the 119/9 strains could not be determined from the assembly. However, the two varieties have highly collinear genomes [31]. We therefore plotted the dS along the gene order of the var. *bisporus* homokaryon H97 genome.

### 2.4. Detection of Centromeres

Centromeres have been identified and discussed previously [31]; in order to place them on our plots, we used three independent lines of evidence based on their features [31]. First, we downloaded transposable element (TE) sequences annotated for the *A. bisporus* var. *bisporus* H97 v3.0 genome [74] and investigated their positions along chromosomes. Second, we investigated the positions of orthologous gene pairs found in H97 and H39 genomes. Lastly, we investigated the 5mC methylation along H97 chromosomes. We considered genomic regions as centromeres following a previous publication [31], i.e., where (1) TEs are found in abundance (i.e., TE islands, (2) methylation is concentrated on TEs in the region and (3) there is a depletion of orthologous genes in the regions. To do these simultaneously, we used circos plots [75] showing TEs, methylation and orthologous genes along H97 and H39 genomes. We used bisulfite sequencing (BSseq) data published previously [76]. We trimmed BSseq data with TrimGalore v. 0.6.5 [77]. We used bismark v. 0.22.3 [78] to map BSseq data to the reference genome (H97 v3.1; [27]) and subsequent extraction of methylation data. We considered a cytosine as methylated only when it has at least 3X coverage, and the majority of the reads remained not converted to uracils.

### 2.5. Linkage Map of Mating-Type Locus of the Variety burnettii

Markers of a previously generated linkage map of the var. *burnettii* strain 119/9 [31] were used to plot genetic to physical distances of a 1 Mb region on chromosome 1 that contains the HD mating type genes. This map was generated with Genotyping by Sequencing data. The physical distance between adjacent markers (kb) was determined and divided by the genetic distances between these markers (cM). The kb/cM data were subsequently plotted to the physical distance (kb). A few markers were removed because they were in the wrong order, thus generating negative values. Wrong ordering of markers can occur especially in dense maps [79].

## 3. Results

### 3.1. Defining the Exact Location of the MAT Locus on Chromosome 1

A new order of mating-type-linked molecular markers assigned to chromosome 1 [38,54] has been established (Figure 1).

This order is statistically as likely as the one previously defined [38], with a similar magnitude of log-likelihoods: the difference between the two log-likelihood values is 0.63, i.e., the likelihood of the new order is 10^0.63^ = 4.26 greater than the older one. The total distance of the linkage group is unchanged. Except for the marker PR057, the positions of the markers are consistent with the expected order according to the genome sequence. Thus, the available genome sequence allowed defining the location of the sequenced marker AbSSR23 (GenBank X92961) to position 1077972-1081182 on chromosome 1 of strain H97 and of the marker ACM5C (GenBank GS376049) to position 831821-832017. The primers defining the CAPS marker PR006 (CAATCTCAAGCTTGCCTGC; AGGTGACATGTCAGAAGCGC; [38]) match on chromosome 1 to positions 862656-862675 (+ strand) and 863836-863854 (− strand) (Figure 1). Another closely linked marker, PR067 (AB126057), locates at position 1097918-1100046 on chromosome 1 (Figure 1).

The *MAT* locus resides in the mapping interval between the marker AbSSR23 and the marker ACM5C. With our marker segregation data, we are unable to map more precisely the *MAT* locus, as the probabilities of having the marker order AbSSR23-*Mat*-PR006-ACM5C or the alternative one, AbSSR23-PR006-*Mat*-ACM5C, were comparable, with a log-likelihood difference between the two orders of ca 0.35. However, the identification of *HD* genes within the same interval (see below) strongly supports the location of the mating-type locus in *A. bisporus*.

### 3.2. Identification and Structure of the Mating-Type Homeodomain Loci

Indeed, tblastn searching with *HD2* protein *b2-1* of *C. cinerea* (GenBank CAA56132) on the JGI page of strain H97 identified in the 214 kb region in between markers ABSSR23 and PR006 on chromosome 1 two putative genes (at positions 890959-892499 and 893233-894863). Their respective products contained a classical three-helical homeodomain motif of 60 amino acids [80] as found in basidiomycete *HD2* mating-type transcription factors (Figure 2A). These two *HD2* genes are arranged in tandem (Figure 3). Upstream of the two genes is another gene (at position 895062-897100; Figure 3) coding for a putative transcription factor with a TALE-homeodomain type [81] with three extra amino acids in between the first two helixes and a poorly conserved DNA-recognition motif as typical for basidiomycete *HD1* domains (Figure 2B). Congruent with this, the b1-1 mating-type protein of *C. cinerea* (GenBank CAA44210) matched in tblastn searches the DNA region encoding the TALE homeodomain.

In the classical situation of a basidiomycete *HD* locus, *HD1* and *HD2* genes are divergently transcribed in pairs [82,83,84]. The H97 *HD1* gene *a1-1* is divergently transcribed to the two *HD2* genes and forms with gene *a2-1* a typical divergently transcribed *HD1*-*HD2* gene pair (Figure 3). Downstream of the *HD1* gene transcribed on the opposite strand (at position 897420-899760) follows the *mip1* gene, coding for a putative mitochondrial endopeptidase (ID 239372), that has high levels of identity (72–74%) to orthologous proteins from other Agaricomycetes including *C. disseminatus* (Q6Y5M6), *C. cinerea* (EAU92790) and *Laccaria bicolor* (EDR15793). In these three species also, *mip1* flanks one side of the *HD* mating-type loci, whereas gene *β-fg* encoding a conserved fungal protein of unknown function flanks the other side [48,85,86]. In *A. bisporus* var. *burnettii* JB137-S8, downstream to the *HD2* genes of the *MAT* locus, encoded on the opposite strand, are two copies of *β-fg* genes (at positions 887425-889480 and 883864-884895), while a single copy has been observed so far flanking *HD* mating-type genes in other Agaricomycetes. The two *β-fg* gene copies are also found in the *A. bisporus* var. *bisporus* H97 strain (as noted in a preliminary analysis [52]) and display several differences. The structure of the shorter gene *β-fg1* with six introns (one after the start codon) and the length and amino-acid sequence of its protein are highly similar to the genes and proteins in *C. cinerea* [87], *C. disseminatus* [48], *L. bicolor* [86], *Pholiota microspora* [88] and *Heterobasidion annosum* [49]. The 220 aa-long product (ID 1137415) of *β-fg1* shows 57–58% identity to the products of *β-fg* genes in *C. disseminatus* (AAZ14920), *C. cinerea* (EAU92782) and *L. bicolor* (EDR15176). *β-fg2* is much longer, with 14 introns (also one after the start codon), due to multiplication of a 5′ intron-exon unit, and encodes a protein of 406 amino acids (ID 1181595) with an N-terminal half composed of 7 1/2 repeats of a 20 amino acid motif (N_8/8_K_7/8_T_6/8_G_7/8_V_6/8_E_7/8_G_8/8_H_6/8_N_8/8_A_5/8_P_6/8_Q_7/7_S/T_5/7_G_7/7_H_6/7_P_7/7_Q_7/7_S/T_7/7_G_5/7_V_6/7_). In *Rhodonia* (*Postia*) *placenta*, the mating-type locus with the homeodomain transcription factor genes is also flanked by a gene encoding an enlarged β-fg protein possessing N-terminal repeats (*R. placenta* protein ID 124109 [89]).

The gene order *mip1*-*HD1* and *HD2*-*β-fg* with an *HD1*-*HD2* gene pair divergently transcribed is known from the *HD* mating-type locus of several other Agaricomycetes [42,48,50,86,90,91]. The gene content and order of the region and its location between molecular markers AbSSR23 and PR006 (Figure 1) firmly linked to mating-type function strongly supports that the single *A. bisporus MAT* locus corresponds to these *HD* genes. The genetic distance between PR067-AbSSR23, AbSSR23-*MAT*, *MAT*-PR006 and PR006-ACM5C are 1.9 cM, 4.3 cM, 1.7 cM and 4.8 cM, respectively. For these same intervals, physical to genetic distances correspond to 8.81, 41.44, 11.77 and 6.38 kb/cM, respectively. In the intervarietal cross *A. bisporus* var. *bisporus* x var. *burnettii*, the recombination rate is on average 26.32 kb/cM [38], and the *MAT* locus thus appears to reside at a region with little recombination [38], similar as the *HD* mating-type locus in *C. cinerea* and other basidiomycetes.

We also searched for potential genes coding for proteins with pheromone activity and with pheromone receptor activity in the H97 genome. Four genes for putative precursors of lipid-modified pheromones (Appendix A) were found by tblastn searching with the sets of mating-type pheromone precursors and the precursor for pheromone-like non-mating-type specific peptides identified from genomes of *C. cinerea*, *L. bicolor* and *Schizophyllum commune* [8,86,87,92]. Three of them localize on chromosome 1 to the regions that flank the two ends of a large island rich in transposable elements (about 270 kb in length), corresponding to the centromere [31], at a chromosomal position that links only loosely with the *MAT* locus (Figure 1; distance > 50 cM Kosambi). The fourth gene is present on chromosome 2 [51] (Appendix A). We found four genes encoding putative STE3-like pheromone receptors (Appendix A) in tblastn searches using the sequences of all putative STE3-like receptors of the *C. cinerea* Okayama 7 genome [87]. One of the four H97 STE3-like genes (*ste3.1*) localizes on chromosome 5 and one (*ste3.4*) on chromosome 12. Two others (*ste3.3*, *ste3.2*) are present on chromosome 13 (Appendix A). The scattering of pheromone precursor genes and pheromone receptor genes over several different scaffolds together with lack of linkage or, in the best case, only loose linkage to *MAT* activity further supports that none of these genes has a mating-type function.

### 3.3. Orthologous Mating-Type Genes in the A. bisporus var. burnettii JB137-S8 Genome

The genes described above in *A. bisporus* var. *bisporus* H97 (Appendix A) were used to screen by blast and tblastn searches the genome of *A. bisporus* var. *burnettii* JB137-S8 on the respective JGI data page. All genes of strain H97 for putative pheromone precursors and for putative pheromone receptors had orthologs in the JB137-S8 genome at homologous chromosomal positions (Appendix A; see below) with products sharing high identity (97–100%) with the H97 gene products (Appendix A).

The *mip1* gene and the two *β-fg* gene copies in JB137-S8 had products showing high identities (98–99%) with the H97 proteins (Appendix A). In sharp contrast, the HD1 and HD2 mating-type proteins were little conserved in sequence, consistent with expectations given their typical high rate of evolution [8,93]. The product of *HD1* gene *a1-2* of strain JB137-S8 revealed only 76% identity to protein a1-1 that comes from the *HD1* gene localized in the H97 locus at the position corresponding to gene *a1-2* in JB137-S8 (Appendix A; Figure 3). At the DNA level, they share an identity of 88% over their whole length, making it very likely that they are alleles of the same *HD1* gene. Strain JB137-S8 has only a single *HD2* gene. *b2-1* corresponds in location to the *HD2* gene *b2-2* in strain H97 (Figure 3), while the two genes do not show significant DNA similarity with the exception of a short stretch of DNA in a region encoding part of the HD2 homeodomain (73% identity). The products of *b2-1* and *b2-2* share, however, 26% identity in protein sequence (Appendix A), suggesting that also these could be alleles of one gene. In contrast, identities (35%) of b2-1 to the other H97 HD2 protein a2-1 were restricted to only a 118 aa-long region with the HD2 homeodomain (Figure 2A). We found no significant similarity at the DNA level between the two genes.

Whereas no further *HD2* gene was found in the JB137-S8 genome, there was a second *HD1* gene in its *MAT* locus. *b1-1* forms with gene *b2-1* a divergently transcribed *HD1* and *HD2* gene pair (Appendix A; Figure 3). The first 231 aa of its product show 33% identity to the region in the H97 HD1 protein a1-1 which covers the HD1 homeodomain (Figure 2B) and, by analogy to other species, likely the N-terminal specificity domain required for heterodimerization with compatible HD2 proteins [43,93]. High similarity (72% DNA identity) between the two genes are restricted to the DNA stretch encoding the conserved helix III of the homeodomain, making it unlikely that *b1-1* is an allele of *a1-1*.

In summary from the comparison of the genes in the *MAT* alleles of the two sequenced strains, it appears that two paralogous *HD1-HD2* gene pairs (a.k.a. *HD* subloci) exist in *A. bisporus*, although in neither of the two *A. bisporus* varieties both gene pairs were complete. Analogous to the paralogous *HD1-HD2* gene pairs in *C. cinerea* [83,94], we called these the *a* pair and the *b* pair (Figure 3). The *A. bisporus* var. *bisporus* strain H97 has a complete *a* pair and misses an *HD1* gene of the *b* pair, and the *A. bisporus* var. *burnettii* strain JB137-S8 has only an *HD1* gene of the *a* pair but a complete *b* pair.

Interestingly, the length of DNA between gene *mip1* and the two *β-fg* genes differs much in length between the two sequenced strains of different varieties (Figure 3). The region in strain H97 is 10 kb long versus ca. 42 kb in strain JB137-S8. A sequence comparison revealed 98% DNA identity for the coding regions of the *mip1* alleles, 97% DNA identity in the downstream 387 bp long intergenic regions, 88% DNA identity for the alleles of the *a1* gene and 77% DNA identity for the 559 bp-long sequence fragments upstream to the start codons of the *a1* alleles (Appendix A). The data support that, as in other species [48,85,90,95], there is a sharp decline in DNA identity from the end of the *mip1* gene (including its terminator region) to the start of the *MAT* locus. Interestingly, the 559 bp-long fragment upstream of the genes *a1-1* and *a1-2* covers the promoter region plus 386 bp of a start of a divergently transcribed *HD2* gene (77% DNA identity to *a2-1*), suggesting that there is a short remnant of a former *HD2* gene *a2-2* in strain JB137-S8 (Appendix A, Figure 3 and Figure 4a). Furthermore, there is a 207 bp sequence in close vicinity in strain JB137-S8 that appears to represent a footprint of the 3′ end of a former *mip1* gene (77% DNA identity to the 3′ end of *mip1* of strain H97; Figure 4B) that had the same direction of transcription than the extant *mip1* gene (Appendix A, Figure 3). It is possible that this *mip1* footprint originated from a duplication of a complete *HD1* and *HD2* gene pair, represented in strain JB137-S8 by the genes *b1-1* and *b2-1* (Figure 3).

We found a long stretch of extra DNA (35 kb estimated in length) in the *MAT* locus of the *A. bisporus* var. *burnettii* strain JB137-S8 between the *mip1* footprint and the *b* gene pair. This 35 kb DNA fragment contains several sets of direct and inverted repeats as well as several corroded sequences for reverse transcriptases, integrases, maturases, gag-pol-proteins, a superfamily 21 transposase and a protein with a hAT family dimerization domain (Figure 3) and should thus originate from multiple insertions of transposable elements into the *MAT* locus.

In the *A. bisporus* var. *bisporus* strain H97 on the other hand, there is a 1.6 kb long insertion in between the 1 kb shared sequences of the intergenic region (97% DNA identity) of the two copies of the highly identical *β-fg* genes (98 and 94% DNA identity between the alleles; Appendix A; Figure 3). This insert is flanked by inverted repeats of 65 bp (Figure 3), and there are other copies in the genomes of the two strains indicating it also to be a transposable element. As in other species [48,85,90], the sharp increase in DNA identity from homeodomain transcription factor genes to the location of the neighboring *β-fg* gene marks the end of the *MAT* locus (Figure 3). The chromosomal region of *A. bisporus* defined here as the *MAT* locus fulfills thus the prerequisite of basidiomycete mating-type loci [82,93] of containing unique DNA sequences with highly diverged alleles of genes with potential functions in control of sexual development and mating compatibility.

### 3.4. MAT Genes and Their Products

EST sequences were available for all three homeodomain transcription factor genes of strain H97 (https://mycocosm.jgi.doe.gov/Agabi_varbisH97_2/Agabi_varbisH97_2.home.html accessed on 27 March 2011) indicating that all three identified mating-type genes are expressed. *a1-1* and *a2-1* both have three introns, and *b2-1* has two. Resulting mature mRNAs encode proteins of the lengths of 627, 464 and 491 aa, respectively (Appendix A). The lengths of the JB137-S8 proteins a1-2, b1-1 and b2-1 are 624, 613 and 455 aa, respectively (Appendix A). As in *C. cinerea* [85] and in *L. bicolor* [86], the HD1 proteins are therefore longer than the *HD2* proteins. C-terminal to the HD1 homeodomains, the proteins contain a putative bipartite nuclear localization signal (NLS; Appendix A) at corresponding positions to NLSs in the *C. cinerea* and *L. bicolor* HD1 proteins [86,96]. The regions C-terminal to the homeodomain in the HD2 proteins have no recognizable domain but in all three proteins the C-terminal halves are serine- and threonine-rich (23–26% of all aa; Appendix A). The *A. bisporus* HD1 proteins have an N-terminal domain of about 107-111 aa of predicted helical structure (Appendix A). The N-termini of the *A. bisporus* HD2 proteins of a2-1 and b2-2 are 152 and 148 aa long, whereas the N-terminus of b2-1 is shorter with 99 aa and apparently N-terminal-truncated (Appendix A). These domains have a predicted helical structure. The corresponding regions in functional *C. cinerea* and *S. commune* HD1 and HD2 proteins are of a similar length (ca. 110 to 120 aa in HD1 proteins; ca. 130-150 aa in HD2 proteins) and predicted structure, and have been shown to confer mating-type specificity by discriminative heterodimerization between HD1 and HD2 proteins [85,97,98].

The *A. bisporus* HD1 proteins as well as the HD2 proteins in blastp searches of the NCBI GenBank hit the respective HD1 and HD2 homeodomains of basidiomycete mating-type proteins and often also their sequences for the N-terminal dimerization domains (in HD1 proteins defined as pfam12731 = Mating_N domain). We used the N-terminal sequences up to the last aa of the homeodomain to perform phylogenetic analyses of allelic and paralogous HD1 and HD2 proteins of basidiomycete mating type proteins (Figure 5). The two orthologous HD1 sequences a1-1 of var. *bisporus* and a1-2 of var. *burnettii* branched together (Figure 5A). As expected, the paralogous HD1 sequence b1-1 was placed farther away, together with orthologous sequences from other species (Figure 5A). The two paralogous HD2 proteins of the *A. bisporus* var. *bisporus* H97 (a2-1 and b2-2) branched together but not with b2-1 of *A. bisporus* var. *burnettii* JB137-S8 (Figure 5B), possibly because of the N-terminal truncation of protein b2-1 that might hinder a more specific clustering of the protein.

### 3.5. Genes for Pheromone-Like Peptide Precursors

The amino-acid sequences deduced from the four genes in each of the two sequenced *A. bisporus* varieties for putative precursors of pheromone-like peptides are given in Figure 6. EST data indicate that at least three of the four precursor genes are expressed, all three with transcripts that contain an intron in the 3′ untranslated region (https://mycocosm.jgi.doe.gov/Agabi_varbisH97_2/Agabi_varbisH97_2.home.html accessed on 2011-04-01).

The four precursors are between 44 and 54 aa long (Figure 6). Although all allelic pheromone precursors but the Phl2 precursors differed by one amino-acid from each other, the mature peptides would be alike between the strains of the two varieties and should thus confer no different pheromonal activity. Pheromone precursors are expected to be N-terminal processed first for secretion by proteolytic cleavage after the outmost conserved MDS/AF motif and then by a peptidase cleaving at a charged dipeptide motif such as ER/DR/EH/NH/EE/DE [8,93,100]. The C-terminal CAAX motif (C = cysteine, a = aliphatic aa, X = any aa) of precursors for lipid-modified fungal pheromones is modified by farnesylation of the cysteine residue, removal of the three terminal amino acids and carboxymethylation of the resulting carboxy-terminal cysteine [99,100]. All detected pheromone precursors possess internal charged amino acid motifs in expected distances to the characteristic C-terminal CAAX motif (Figure 6). The predicted four mature peptides of a strain are about 11–13 aa long and differ from each other in sequence by one or more aa (46–91% identity/similarity). All mature pheromones are G-rich (Figure 6), and three of them resemble non-mating-type pheromone-like peptide motifs of *C. cinerea* (consensus GGGSGNGAYC) and *L. bicolor* (consensus GGGSGNNAYC) [8,86].

### 3.6. Genes for Pheromone Receptors

Analyses with the TMpred and the SPLIT 4.0 servers supported for all four predicted *Agaricus* STE3-like pheromone receptors a seven-transmembrane structure, while the TMHMM Server v. 2.0 predicted with good probability for STE3.4 only six transmembrane helices (the third helix being not well supported), but for the pheromone receptors STE3.1, STE3.2 and STE3.3 also seven. The sequences for the seven-transmembrane helices were used with those from proteins of *C. cinerea*, *L. bicolor*, *R. placenta* and *S. commune* to generate a phylogenetic tree (Figure 7). In *C. cinerea*, three paralogous subgroups of a pheromone receptor gene plus pheromone precursor genes in the *PR* mating-type locus were functionally defined [101,102,103]. The positions of the pheromone receptor genes within the *PR* mating-type locus and at non-mating type chromosomal sites have been previously found reflected in trees with proteins of *C. cinerea*, *L. bicolor* and *R. placenta* [8,86,89,104,105], which is confirmed in this study (Figure 7). Of the *Agaricus* proteins, only the STE3.2 receptors group with proteins by mating type, all belonging to subgroup 2 of mating-type pheromone receptors (Figure 7). In contrast, the STE3.3 genes and the STE3.1 and STE3.4 genes cluster with two distinct groups of non-mating type pheromone receptors encoded by two genes found in all four other species in close distance to the *PR* mating-type loci [86,87,89,92]. In *C. cinerea*, these are the genes for CcSTE3-2151 and CcSTE-2153 [86,87].

The tree thus supports the inference from previous comparative analyses [87] that the ortholog of the pheromone receptor gene *CcSTE3.2a*/*rcb2* of the *C. cinerea PR* mating-type locus is in *A. bisporus* on the chromosome 13. From 105 analysed genes from chromosome 13 of *A. bisporus* var. *bisporus* H97, 71 had related genes on chromosome 10 of *C. cinerea* (Appendix A). Importantly, the gene *ste3.2* sits in a block of four genes and gene *ste3.3* in a block of five genes showing a conserved chromosomal order to the respective *C. cinerea* gene blocks including the *PR* mating-type receptor gene *CcSTE3.2a*/*rcb2* and the non-mating-type pheromone receptor gene *CcSTE-2153*, respectively. These data and the tree shown in Figure 7 strongly suggest that the location of the locus orthologous to the *PR* mating-type locus has been found on the chromosome 13 in *A. bisporus* (*ste3.2*).

### 3.7. Differentiation along Mating-Type Chromosomes

We analysed the differentiation along mating-type chromosomes by plotting the per-gene synonymous divergence in the heterokaryotic strains (dS, Figure 8). The dS level is expected to increase with time because of the linkage with the mating-type locus, as the cessation of recombination allows the accumulation of different substitutions in the alleles associated with the different mating-type alleles [11]. In the cultivated U1 strain *A. bisporus* var. *bisporus*, we found high dS levels along most chromosomes and including edges of chromosomes (Figure 8A, Appendix A). Unexpectedly, we found some genomic regions with zero dS within some chromosomes, not extending up to edges (Figure 8A), indicating that the two parents used to generate the cultivated strain were related, as previously suggested [27]. Selfing in this variety with recombination being restricted at chromosome ends would indeed not generate such islands of homozygosity in the middle of chromosomes. The dS pattern in the cultivated U1 strain *A. bisporus* var. *bisporus* is thus consistent with this variety resulting from an inbreeding event and being clonally replicated for cultivation [27]. The very high differentiation between the *β-fg1* alleles (Figure 8A), much higher than the 99th percentile values of the dS distribution (Appendix A), indicates an ancient event of recombination suppression at the mating-type locus and encompassing the *β-fg1* gene, which is consistent with its linkage to *HD* genes in most Agaricomycota. The alleles at the *HD* genes were also highly differentiated between mating types, much higher than the 95^th^ percentile values of the dS distribution (Appendix A), as expected given their role in mating-type determinism (Figure 8A). The alleles at the *mip* gene were less differentiated, indicating more recent complete linkage to the mating-type *HD* genes. An even more recent event of recombination suppression occurred in *A. bisporus* var. *bisporus*, suppressing recombination across whole chromosomes, as indicated by linkage map analyses [31,38]. This event is more recent as it is found only in the variety *bisporus* and maybe only in the cultivated strain. As expected given their lack of role in mating-type determinism, the STE3-like genes were not particularly highly differentiated between mating types (Figure 8A).

For *A. bisporus* var. *burnettii*, we studied dS levels in the heterokaryotic strain called 119/9, because the genomes of the constituent homokaryotic strains (H119p1 and H119p4) were available. In these strains, we also detected two copies of *β-fg* genes, the *mip1* gene as well as an *HD1-HD2* pair (Figure 9). We could not assemble the mating-type locus in a single contig in the H119p1 genome. We could nevertheless detect a ca. 140 kb-long inversion between mating types, trapping the *HD* genes, the *mip1* gene and fragments of about 130 kb of additional genes, further supporting recombination suppression in several successive steps. We did not find evidence of any additional *HD* gene in this 119/9 heterokaryotic strain. One breakpoint of the inversion was placed between the *HD1*-*HD2* gene pair and the gene *β-fg2* and the other breakpoint within a conserved Agaricomycotina gene of unidentified function, splitting the gene coding for Agabi119p4_1168 in strain H119p4 into *g2660* and *g4769* in strain H119p1 (Figure 9).

We plotted the synonymous differentiation between H119p1 and H119p4 alleles along the well-assembled H97 genome and found high dS levels along most chromosomes (Figure 8B, Appendix A), as expected in an outcrossing species. Some regions with zero dS were nevertheless detected in some parts of chromosomes including edges. The dS plot thus seems mosaic as in the var. *bisporus*, with regions of high dS and regions of zero dS, but with an opposite pattern: the zero dS regions are at the edges of the chromosomes in the var. *burnettii*, while they are more in the middle of chromosomes in the var. *bisporus*. The zero dS regions extending up to chromosome edges in the var. *burnettii* are expected after a few selfing or inbreeding events when recombination is not restricted along chromosomes. Selfing may have occurred in the laboratory before sequencing this heterokaryotic strain [31]. The mating-type chromosome seemed to be the only one to have retained heterozygosity along its whole chromosome. Strikingly, the limit zero/non zero dS seemed often to occur at STE3-like genes (Figure 8B), although we cannot see any reason for such a pattern. The alleles at the *HD* genes were highly differentiated, but also the alleles of the *β-fg2* gene, further supporting an extension of recombination suppression in several steps.

### 3.8. Recombination Suppression at the Mating-Type Locus in A. bisporus var. burnettii

Recombination suppression across whole chromosomes was clearly shown from a previous linkage map analysis in the U1 strain of *A. bisporus* var. *bisporus* except chromosome edges [31]. Recombination was more evenly distributed along chromosomes in *A. bisporus* var. *burnetti*, but the previous publication did not report finely recombination across the mating-type chromosome in this variety. We, therefore, re-analysed here the data focusing on the mating-type locus region. Markers of a 1 MB region of this map containing the *HD* mating type genes were used to plot the genetic to physical distance, showing a clearly higher genetic distance between the markers flanking the *HD* genes than between other marker pairs. That indicates a strong recombination reduction in the mating-type locus region (Figure 10).

## 4. Discussion

A single mating-type locus governs mating compatibility in the button mushroom *A. bisporus*. Using linkage analysis with a genetic map, we could assign the *MAT* locus on chromosome 1 to a DNA region that contains *HD1* and *HD2* homeodomain transcription factor genes and that is flanked by the *mip1* gene, corresponding to a putative mitochondrial endopeptidase, and copies of the *β-fg* gene, with no predicted function, as described for several *HD* mating-type loci in Agaricomycetes [8,46,48,50]. *A. bisporus* is thus another case reported in the Agaricomycetes where bipolarity is found caused by the loss of mating-type function by the pheromone-pheromone receptor system at the *PR* locus, after *C. disseminatus* [48], *P. microspora* [46,50], *Phanerochaete chrysosporium* [47], *Volvariella volvacea* [106] and the antrodia clade of the Polyporales with *R. placenta* and *Wolfiporia cocos* [89,107]. Analyses of multiple Agaricomycetes supported that tetrapolarity is the ancient situation and bipolarity the derived one [107]. In other basidiomycete clades, such as in the Tremellomycetes, Trichosporonales, Microbotryales and Ustilaginales, bipolarity has also repeatedly evolved, but this occurred through chromosomal rearrangements and linkage by recombination suppression of the two *MAT* loci rather than the loss of the mating-type role by the *PR* locus [8,10,12,14,44,106,108,109,110].

The high genetic differentiation between alleles at the *β-fg* genes at the mating-type locus together with the presence of an *β-fg* gene near the *HD* genes across Agaricomycetes [86] altogether indicate an ancient recombination suppression event trapping these genes at the mating-type locus. The inversion at the mating-type locus in the 119/9 heterokaryotic *A. bisporus* var. *burnettii* strain trapped the *mip1* gene, while the differentiation between alleles at the *mip1* gene was low, indicating another, more recent, recombination suppression event. We found duplications of the *β-fg* gene, as noted before [52], and, in addition, footprints of two *HD* subloci but with the deletion of one *HD* gene in one pair, although not the same in the reference genomes of the two varieties. Such gene duplications and losses are also a hallmark of regions with ancient recombination suppression [111,112]. Such an extension of recombination suppression to a few other genes near mating-type genes has also been reported in other fungal clades, with different genes involved depending on the clades [11]. The linkage of these genes to mating-type genes may sometimes be adaptive, as in the case of genes involved in mitochondria inheritance when mitochondria are transmitted by a single mating-type, and the *mip1* gene may be involved in this phenomenon [113]. However, in most cases, as for the *β-fg* gene in *Agaricus*, it is not clear whether and why it would be beneficial to link different alleles to the mating-type genes; it has therefore been proposed that the extension of recombination suppression occurred for other evolutionary causes than antagonistic selection, such as for sheltering deleterious mutations or by the spread of transposable element methylation marks that accumulate in non-recombining regions [11]. In multiple fungi, the recombination suppression has extended across large regions distal to the mating-type genes without any genes involved in mating-type determinism, which seems to occur particularly often in pseudo-homothallic fungi [10,14,15]. Recombination suppression has extended across the whole mating-type chromosomes in the cultivated strain of the pseudo-homothallic *A. bisporus* var. *bisporus* variety and intriguingly also across whole autosomes [27,31]. Recombination suppression has thus extended stepwise, with at least two steps, in the *A. bisporus* var. *bisporus* mating-type chromosome, with first the *β-fg* gene given its high genetic differentiation, and then across the whole chromosome, as indicated by linkage map analyses. Such stepwise extension of recombination suppression fits the general pattern in pseudo-homothallic fungi [10,11,14,15,114]. As reported in other ancient non-recombining regions around mating-type genes [10,115], transposable elements have accumulated near the *HD*, *β-fg* and *mip1* genes in the reference genomes of the two *A. bisporus* varieties. The evolutionary and proximal causes of such stepwise recombination suppression are still debated [11], and the *A. bisporus* var. *bisporus* case adds a nice biological model for testing hypotheses. It will be interesting to assess whether recombination suppression across whole chromosomes occurred only in the cultivated strain or occurs in the whole *A. bisporus* var. *bisporus* variety. Another sequenced commercial var. *bisporus* strain had just a simple *MAT* locus restricted to only a single 4 kb-long *HD1-HD2* gene pair, comparable to the H97 *a* pair [52]. Crosses between homokaryons of the U1 cultivar and wild *A. bisporus* var. *bisporus* homokaryons still displayed recombination restricted to chromosome ends [116], but recombination patterns have not been studied in wild strains of this variety. Stepwise recombination suppression seems to have occurred in several fungal mating-type chromosomes [11,14,114], as in many sex chromosomes [117]. The reasons why are still not understood [11], but several hypotheses have been invoked, such as the sheltering of deleterious mutations through permanent heterozygosity [118]. In the case of the U1 cultivar, the final step of recombination suppression along all chromosomes, and not only the mating-type chromosome, may alternatively be due to a strong selection under domestication for propagating the same heterozygous genotype without recombination, just as selfing annual crops or fruit tree crops propagated by grafting.

In the reference genomes of the two *A. bisporus* varieties, we found the genes orthologous to the *PR* mating-type locus scattered across the genome and lacking a mating-type role. The pheromone receptor gene *ste3.2* orthologous to *PR* mating-type pheromone receptor genes in other Agaricomycetes locates on chromosome 13, and no pheromone precursor gene was found linked to this gene. The alleles at these genes were not differentiated between mating types, while in basidiomycetes the *PR* alleles usually show relatively high differentiation due to very ancient recombination suppression [119,120]. As in *A. bisporus*, *PR* genes do not control mating type anymore; pheromone precursor and receptor genes do not need to be physically linked or differentiated between mating types, while they could still perform important functions in the sexual cycle. Four pheromone-precursor-like genes were detected in the *Agaricus* genomes that relate to multiple similar genes in *C. cinerea* and *L. bicolor*, and they could activate the pheromone receptor STE3.2 independently of mating. Alternatively, mutations in the *PR* pheromone receptor gene could set active the downstream signalling system independently of mating. Indeed, in *C. cinerea*, self-compatible STE3 receptors originating from the *B6* mating type have been described that by a single amino acid change (Q229P in the sixth transmembrane helix of rbc2^6^), they constitutively activate the pheromone signalling pathway [100,121]. Another self-compatible mutant receptor (of rcb3^6^) had a R96H substitution at the junction of the third transmembrane domain and the second intracellular loop that caused a change in specificity and allowed the binding of a self-pheromone, i.e., a pheromone encoded in the same *PR* mating-type locus [100,122]. Amino acid V183 from the third intracellular loop was lost in the rbc2^43^ receptor in a self-compatible *C. cinerea* mutant [100]. Loss of two amino-acids (K75L76) in the third transmembrane helix of the *PR* mating-type receptor bbr2 of *S. commune* changed the spectrum of pheromone recognition [123].

Although non-mating type STE3-like receptor genes are broadly found in Agaricomycetes close to *PR* mating-type genes [47,86,87,89,90,92,104], their function and whether they interact with pheromones are still unknown. In the Tremellomycete *Cryptococcus neoformans*, a constitutively active non-mating-type receptor (CPR2) has been characterized that competes with the mating type-specific Ste3α pheromone receptor for signalling [110]. In *A. bisporus*, one of the non-mating type *ste3* receptor genes discovered in the genome still remained close to the proposed ortholog of the *PR* mating-type locus (*ste3.3*), whereas a second gene has likely been translocated from the neighbourhood of the former mating type locus into two other places (*ste3.1* and *ste3.4*), onto two different chromosomes and near the centromere for *ste3.4*. This might be part of a still ongoing degeneration process following loss of function but could also indicate that these genes have an important yet unknown cellular function that needs to be retained. Recent studies of natural expression and overexpression of non-mating type pheromone receptor-like genes in *S. commune* have suggested possible roles in vegetative filamentous growth and/or self-recognition [124].

In the tetrapolar Agaricomycetes species, the *HD* and the *PR* mating-type genes control different parts of the sexual cycle [43,93]. The *HD* locus encodes two types of transcription factors that act upon heterodimerisation as master regulators directly on genes in the nucleus [43,93]. Interactions between pheromones and their receptors start a signalling pathway that ultimately will target at regulating genes in the nucleus [82], including a kind of epistatic fine-tuning of *HD*-regulated genes that can explain the *PR*-improved frequency of *HD*-induced fruiting in *C. cinerea* [125]. In bipolar species with the *PR* mating-type locus not controlling mating type anymore, such fine-tuning, if still occurring, may follow another way. Overexpression of *HD* genes in the bipolar *P. microspora* caused clamp cell fusion [88], being in tetrapolar species a *PR*-controlled phenotype [43,93]. Clamp cell fusion occurs only after *HD*-induced clamp cell formation [93]. Therefore, the *HD* mating-type genes may well be involved in the molecular process occurring within the cells due to *PR* mating-type function, and the visible *PR*-regulated phenotype of clamp cell fusion could be mediated via the *HD* mating-type proteins. In the bipolar *P. chrysosporium* lacking also a functional mating type-determined pheromone-pheromone receptor-signalling system, nuclear migration has been observed between mating colonies [47], a phenotype controlled in tetrapolar species by the *PR* mating-type genes also in absence of compatible *HD* mating-type proteins [43,93]. Confrontation tests of different *P. chrysosporium* isolates indicated that the *HD* genes at the *MAT* locus can have a positive influence on nuclear migration, but the positive *MAT* control on nuclear migration is not exclusive and never observed to be bidirectional. Not all different *MAT* combinations resulted in nuclear migration, and unilateral nuclear migration was sometimes also observed between strains with identical *MAT* alleles [47]. Numerous mutant “modifier” genes outside of the *PR* mating-type locus in the tetrapolar *S. commune* were found to influence unilateral nuclear migration [126] which offers one explanation as how to bypass the control by *PR* mating-type genes in nuclear migration [47]. Another is that a kind of coordination is still exerted by the still available non-mating-type-specific pheromone and pheromone receptors in the pheromone response pathway [47]. It is conceivable that the *HD* genes in the single *MAT* locus take a positive influence on nuclear migration in *P. chrysosporium* and other bipolar species with observed mating-type effects on nuclear migration [47] by modelling the expression of pheromone and pheromone receptor genes and the pheromone response pathway. Any type of molecular communication between *HD* and *PR* genes has not yet been demonstrated in Agaricomycetes but is well established in the tetrapolar basidiomycete maize smut *Ustilago maydis* from the Ustilaginomycotina [127]. Nuclear pairing in mycelium of tetrapolar Agaricomycetes is *HD* gene-controlled [93,100]. Such control could be sufficient for consorted postmeiotic nuclear migration from basidia into spores of the pseudo-homothallic var. *bisporus* [26]. For the heterothallic var. *burnettii*, nuclear migration between mating homokaryons is rare [26] in accordance with a missing *PR* control.

An important question to address in evolution from tetrapolarity to bipolarity by loss of *PR* mating type function is therefore: are there phenotypes related to mating and/or sexual development that are under sole control of the *PR* genes? There is no nuclear migration or nuclear pairing in the vegetative mycelial cells in *A. bisporus*, and there are no clamp cells at hyphal septa of need to be fused [22,26]. Crucial for the success of mating and sexual reproduction are nevertheless heterologous nuclear recognition and maintenance of the heterokaryotic stage (heterokaryon stability). In the *A. bisporus* var. *bisporus* pseudo-homothallic variety, nuclear recognition at basidiospore formation and/or appropriate positioning of the four haploid nuclei after meiosis ensure that nuclei of different mating types migrate in pairs together into one of the two spores of a basidium [26]. Formation of bisporic basidia has been suggested to be under the control of a main locus, *BSN*, on chromosome 1 but distinct from the *MAT* genes [40,41]. When such basidiospores germinate or when in the heterothallic variety two mating-compatible strains fuse, nuclei of different mating types stay together, although they do not show such tight pairing as it is seen in typical Agaricomycete dikaryons [22,26]. In the typical dikaryotic Agaricomycetes, nuclear pairing and synchronized nuclear divisions occur under regulation of the *HD* mating type genes [128]. As pointed out for *S. commune* [129], nuclear positioning and heterokaryon stability can be further enforced by the *PR* mating-type genes. Loss of mating-type function of the *PR* genes might thus contribute to the random distribution of nuclei in *Agaricus* suggesting the absence of pheromone-mediated nuclear communication. A more relaxed nuclear recognition in the hyphal cells might be antagonized by producing a high number of nuclei per cells, making it statistically less likely that upon cellular divisions one type of nuclei will be fully lost [22]. Under such conditions it could be sufficient for heterokaryon maintenance and sexual reproduction just to retain the *HD* genes’ functionality within a *MAT* locus.

## 5. Conclusions

In this study, we refined the exact location of the mating-type locus on chromosome 1 of *A. bisporus* and report the structure of the *HD* genes and their organization at the mating-type locus for the reference strains of the two varieties *bisporus* and *burnettii*. We found interesting similarities and differences with the *HD* mating-type locus of other Agaricomycetes, and with striking islands of transposable elements. We also identified the gene structure and locus of a pheromone receptor gene orthologous to the *PR* mating-type locus in other Agaricomycetes but having lost its role in mating-type determinism in *A. bisporus*. The high differentiation between the *β-fg* alleles near the *HD* mating-type genes indicates an ancient event of recombination suppression, followed more recently by a suppression of recombination by an inversion trapping the *mip1* gene in at least a strain of *A. bisporus* var. *burnettii* and across whole chromosomes in at least the U1 *A. bisporus* var. *bisporus* cultivar, thus constituting stepwise evolution of recombination suppression. We found evidence of gene duplication and gene losses around the mating-type locus, which is also a hallmark of regions with ancient recombination suppression. Overall, this study thus brings insights into mating-type locus evolution and mating-type gene structures, and can have applied interest for button mushroom breeding.

## Figures and Tables

**Figure 1 genes-12-01079-f001:**
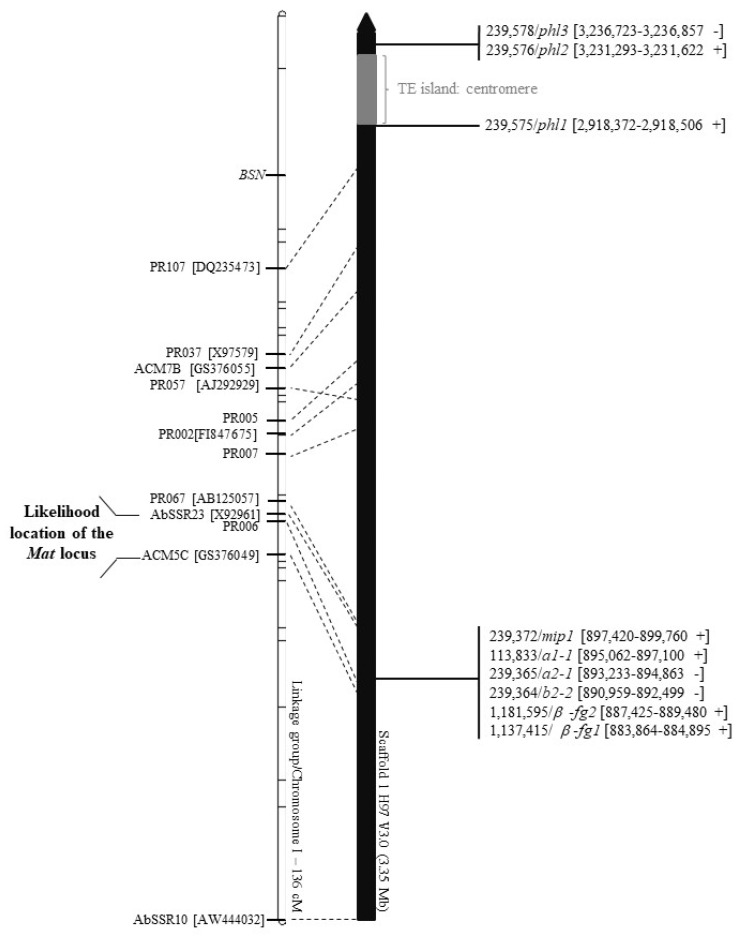
Mapping the mating-type (*MAT*) locus on chromosome 1 of *A. bisporus* var. *bisporus* strain H97 with genetic data on markers previously published [38,54]. Only markers that made possible to anchor the scaffold 1 sequence are located. PRxxx, ACMxx are CAPS (cleaved amplified polymorphism sequence) markers; AbSSRxx are microsatellites. When available, the GenBank reference number is given in bracket. The position of the *BSN* locus controlling the presence of two or four spores in basidia is also indicated.

**Figure 2 genes-12-01079-f002:**
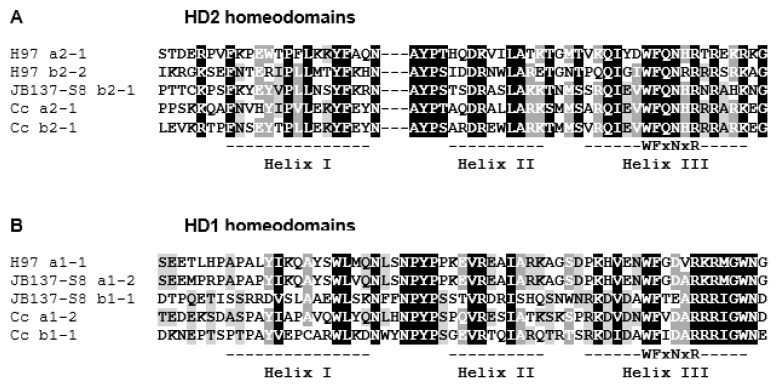
Sequence alignment of (**A**) HD2 and (**B**) HD1 homeodomain motifs of *A. bisporus* mating-type proteins compared to HD1 and HD2 mating type proteins of *C. cinerea* (GenBank EAU92789, CAA44210, CAA566131, CAA56132). Marked are the positions of the three helixes of the homeodomains and the conserved aa in the DNA-binding motif in Helix III (after [80]). Dashes in the HD2 sequences denote the position of insertion of the three extra amino-acid in the HD1 sequences.

**Figure 3 genes-12-01079-f003:**
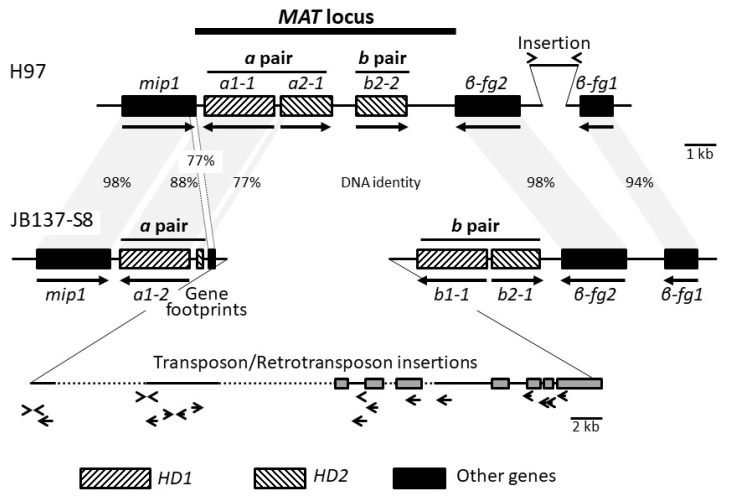
Comparison of the region with the *MAT* locus on chromosome 1 of *A. bisporus* var. *bisporus* H97 and *A. bisporus* var. *burnettii* JB137-S8. Shared genes are shown by black boxes, and *HD1* and *HD2* genes by boxes striped from bottom left to top right and from top left to bottom right, respectively. Closed arrows beneath genes indicate direction of transcription. Genes of DNA similarity or high identity between the two strains are marked with percentages of identity. Insertions are shown by the insets (note the different relative size of the inset in strain JB137-S8). Direct or inversely repeated sequences are marked by arrowheads, and grey boxes indicate DNA sequences that in blastx searches in the NCBI GenBank hit proteins with transposon/retrotransposon activities.

**Figure 4 genes-12-01079-f004:**
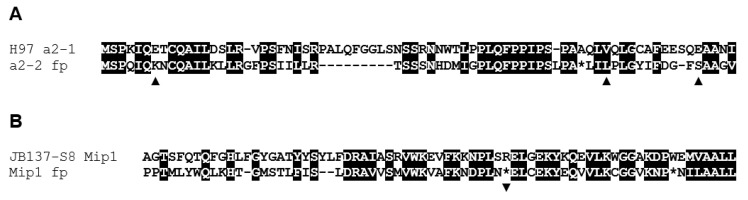
Comparison of (**A**) the first 74 aa of *HD2* protein a2-1 of *A.*
*bisporus* var. *bisporus* H97 with a translated sequence from a footprint (fp) of gene *a2-2* in *A. bisporus* var. *burnettii* JB137-S8 (a2-2 fp) and (**B**) of 71 aa from the C- terminus of Mip1 of strain JB137-S8 and the translated sequence of a *mip1* footprint (Mip1 fp) in the *MAT* locus of the strain (dashes in the sequences are introduced for optimization of the alignment; * mark positions of no aa due to stop codons, ▲ and ▼, beneath the figure positions of frameshifts by insertion or deletion of a base, respectively).

**Figure 5 genes-12-01079-f005:**
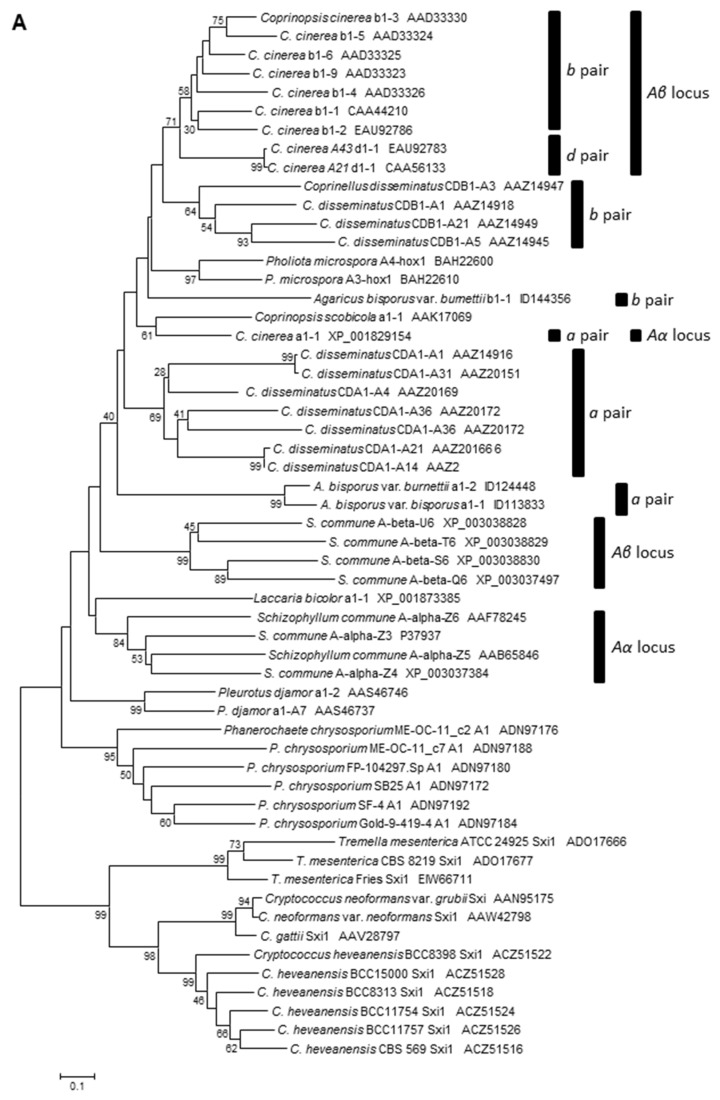
Phylogenetic analysis of the N-terminal dimerization domains and the homeodomains of (**A**) *HD1* and (**B**) *HD2* proteins from Agaricomycotina. Bars mark paralogous proteins from fungi with more than one *HD1*-*HD2* gene pair. Proteins with N-terminal truncations (*A.*
*bisporus* var. *bisporus* H97 b2-1, *Tremella mesenterica* Sxi2) in the ClustalW alignment were manually adjusted to assign them appropriately in position. Extra aa were added to the short N-terminal sequence of *C.*
*cinerea* a2-2 from EAU92788 since the non-functional gene contains a preliminary stop codon in the 5′ end of the gene shortly after the normal ATG start codon [85]. Proteins from Tremellomycetes served as outgroup. Accession numbers refer either to NCBI GenBank or to protein IDs at the fungal JGI databases.

**Figure 6 genes-12-01079-f006:**
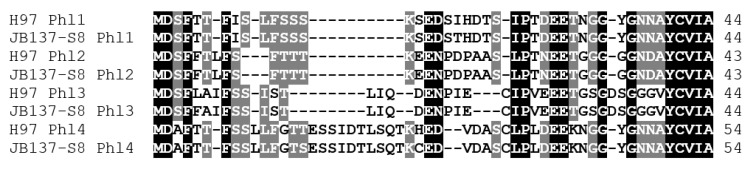
Sequence alignment of pheromone precursors of *A. bisporus* var. *bisporus* H97 and *A. bisporus* var. *burnettii* JB137-S8. Stretches of charged amino acids for possible N-terminal processing [8,93] and the typical CAAX motif at the C-terminus for farnesylation and carboxymethylation [99] are underlined.

**Figure 7 genes-12-01079-f007:**
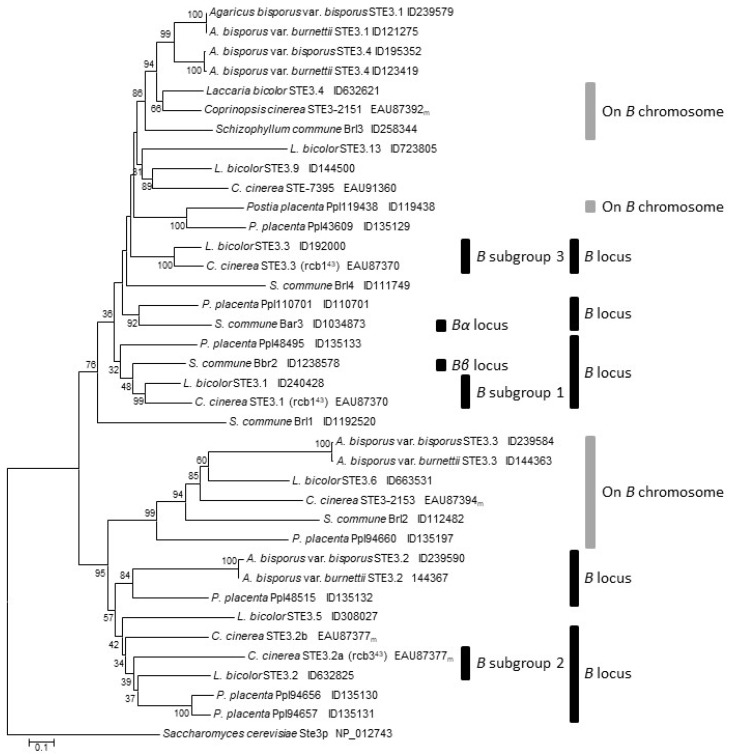
Phylogenetic analysis of the seven-transmembrane regions of putative mating-type-specific and non-mating-type STE3-like pheromone receptors from the Agaricomycetes *A.*
*bisporus* var. *bisporus* H97, *A. bisporus* var. *burnettii* JB137-S8, *C.*
*cinerea* [87], *L. bicolor* [86], *R.*
*placenta* [89] and *S. commune* [92]. The α mating type pheromone receptor Ste3p from the ascomycete *Saccharomyces cerevisiae* served as an outgroup. Black bars mark proteins whose genes are localized within *PR* mating type loci (in the bipolar *P. placenta*, the orthologous *PR* locus with pheromone precursor and pheromone receptor genes seem to have lost mating function [89]. *Bα* and *Bβ* locus and *B* subgroup 1 to 3 indicate paralogous mating type proteins within a species as defined by classical genetics and transformation analyses [8,93]. Grey bars mark genes that are present on the same chromosome than the *PR* mating-type locus. Accession numbers refer either to NCBI GenBank or to protein IDs at the fungal JGI databases. _m_ denotes adjustments done to *C. cinerea* gene models [8].

**Figure 8 genes-12-01079-f008:**
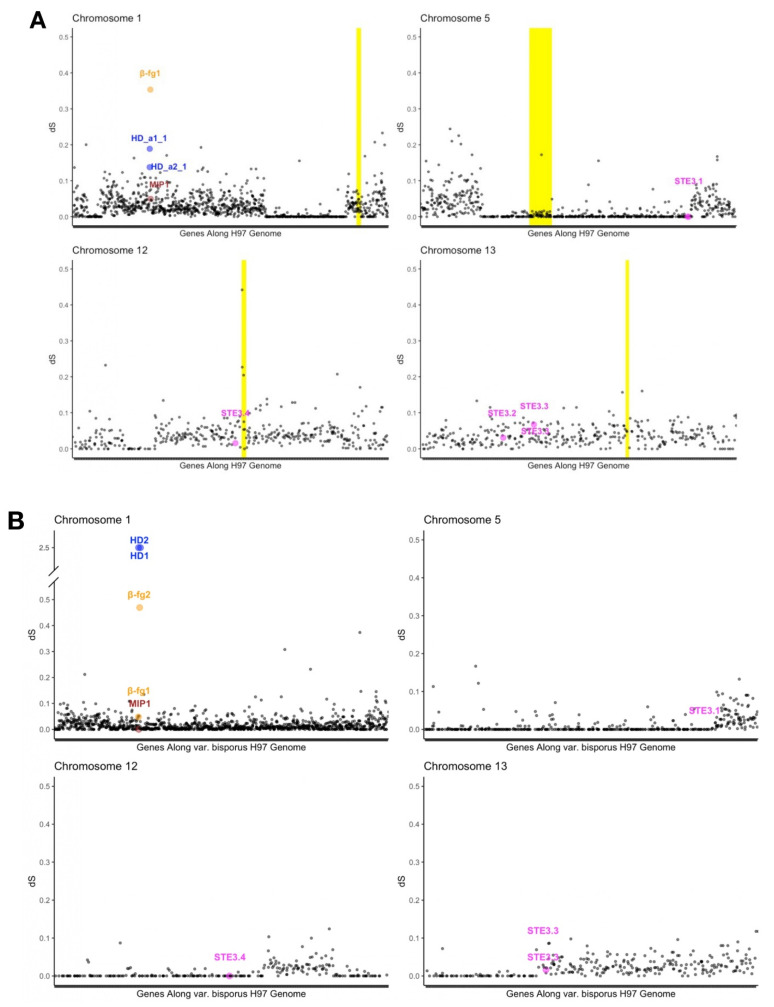
Synonymous substitution rates between the homokaryotic genomes of *A. bisporus* var. *bisporus* H97 and H39 (**A**) and between *A. bisporus* var. *burnettii* H119p1 and H119p4 (**B**). The Y-axis shows the synonymous divergence levels between the alleles found in the homokaryotic genomes of H97 and H39 of alternative mating types, issued from the U1 cultivar. The X-axis shows the gene order along the H97 genome. Yellow highlights on the background show the putative centromeric regions. Genes found in and around the mating-type loci are highlighted with different colours (blue for *HD* genes, orange for *β-fg1* gene, brown for *mip1* gene and magenta for *STE3*-like putative pheromone receptor genes). Other chromosomes are shown in Appendix A.

**Figure 9 genes-12-01079-f009:**
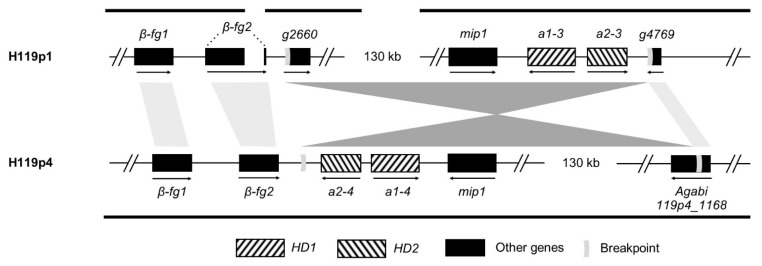
Identity and organization of mating-type genes in the *A. bisporus* var. *burnettii* 119/9 genome. The two constitutive homokaryon genomes (i.e., H119p1 and H119p4) are shown in the top and bottom, respectively. Homeodomain (*HD*) genes are shown with striped boxes, and flanking *β-fg* and *mip1* genes are shown with solid black boxes. Orthologous gene pairs are linked with grey shaded rectangles. The mating-type genes as well as flanking genes are found in three distinct contigs in the homokaryon H119p1 genome (indicated by black bars, the *β-fg2* gene being split onto two contigs), while they are all found *(β-fg* genes in tandem) in one contig in the H119p4 genome. A ca. 140 kb-long inversion was identified between the two genomes of different mating types, figured by the pair of dark grey triangles. The inversion breakpoints are indicated by light grey symbols within the involved genes and non-coding DNA regions.

**Figure 10 genes-12-01079-f010:**
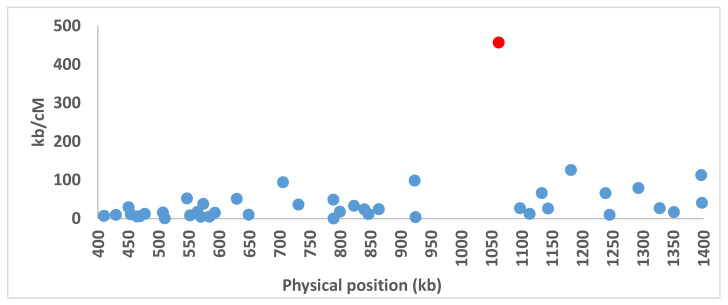
Genetic to physical distance plot for adjacent markers for a 1 MB genomic region on chromosome 1 of *A. bisporus* var. *burnettii* strain 119/9. The red dot indicates the position of the marker pair flanking the *HD* genes on chromosome 1.

## Data Availability

No new genome has been sequenced here.

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
