# Peer review of "Mating-Type Locus Organization and Mating-Type Chromosome Differentiation in the Bipolar Edible Button Mushroom Agaricus bisporus"

_genes, 2021, doi:10.3390/genes12071079_

Round 1

Reviewer 1 Report

Dear Authors, find in included PDF file my comments on the manuscript.

Perhaps some attention might be given in the discussion to a recent article by the last author of the manuscript (https://doi.org/10.1101/2021.05.17.444504).

With kind regards, the Reviewer.

Author Response

Refere 1:

Dear Authors, find in included PDF file my comments on the manuscript.

>> We have incorporated the suggestions from the PDF.

Perhaps some attention might be given in the discussion to a recent article by the last author of the manuscript (https://doi.org/10.1101/2021.05.17.444504).

>> We have incorporated the reference in the discussion.

Reviewer 2 Report

In this paper, Foulongne-Oriol et al describe the organization of the mating-type locus and its differentiation in Agaricus bisporus. The authors present a very detailed molecular examination of the mating-type locus, which will be of interest to researchers in fungal biology and the evolution of the sex chromosomes/loci.

In general, the manuscript is quite long and detailed. The authors should consider moving some of the figures to the supplement to make it more concise and focused on the most critical details being presented. In particular, Figure 1 and the amino acid alignments could potentially be presented as supplementary figures. 

The authors list details of centromere detection in the methods. However, the centromeres are not presented or described in detail in the results text, and are only highlighted in Figure 8. If the identification of the centromeres is novel here, it would be good to see the details of those sequences and how they were found in the results.

In Results section 3.1, the set up for the genetic map is unclear. Although more detail is provided in the methods, it would be helpful for the reader if the logic of this was presented more clearly in this section. 

For Figure 7, S. cerevisiae was used as an outgroup, but the tree is not rooted on S. cerevisiae. The authors should root the tree on the outgroup to make the tree easier to interpret. 

Some of the software versions are missing from the methods section. The authors should add in the version numbers for all software used in the study. 

Author Response

Refere 2:

In this paper, Foulongne-Oriol et al describe the organization of the mating-type locus and its differentiation in Agaricus bisporus. The authors present a very detailed molecular examination of the mating-type locus, which will be of interest to researchers in fungal biology and the evolution of the sex chromosomes/loci.

In general, the manuscript is quite long and detailed. The authors should consider moving some of the figures to the supplement to make it more concise and focused on the most critical details being presented. In particular, Figure 1 and the amino acid alignments could potentially be presented as supplementary figures. 

>> We think that the Figures and the description of the mating-type genes and locus are important for several teams and can always be skipped by readers in case they are more interested in other aspects of the manuscript. We would therefore like to keep them if there is no limitation by the journal.

The authors list details of centromere detection in the methods. However, the centromeres are not presented or described in detail in the results text, and are only highlighted in Figure 8. If the identification of the centromeres is novel here, it would be good to see the details of those sequences and how they were found in the results.

>> We have clarified in M&M that the centromeres have been identified and discussed in a previous paper (Sonnenberg et al 2020).

In Results section 3.1, the set up for the genetic map is unclear. Although more detail is provided in the methods, it would be helpful for the reader if the logic of this was presented more clearly in this section. 

>> We have clarified this section in M&M and results.

For Figure 7, S. cerevisiae was used as an outgroup, but the tree is not rooted on S. cerevisiae. The authors should root the tree on the outgroup to make the tree easier to interpret. 

>> Thanks for the good suggestion, we have rooted the tree with S. cerevisiae.

Some of the software versions are missing from the methods section. The authors should add in the version numbers for all software used in the study.

>> Done.